# *Yersinia* Outer Membrane Vesicles as Potential Vaccine Candidates in Protecting against Plague

**DOI:** 10.3390/biom10121694

**Published:** 2020-12-18

**Authors:** Andrey A. Byvalov, Ilya V. Konyshev, Vladimir N. Uversky, Svetlana V. Dentovskaya, Andrey P. Anisimov

**Affiliations:** 1Komi Research Center, Laboratory of Microbial Physiology, Institute of Physiology, Ural Branch, Russian Academy of Sciences, 167982 Syktyvkar, Russia; konyshevil@yandex.ru; 2Department of Biotechnology, Vyatka State University, 610000 Kirov, Russia; 3Department of Molecular Medicine and USF Health Byrd Alzheimer’s Research Institute, Morsani College of Medicine, University of South Florida, Tampa, FL 33612, USA; 4Research Center for Molecular Mechanisms of Aging and Age-Related Diseases, Moscow Institute of Physics and Technology, 141700 Dolgoprudny, Russia; 5Laboratory for Plague Microbiology, Especially Dangerous Infections Department, State Research Center for Applied Microbiology and Biotechnology, 142279 Obolensk, Russia; dentovskaya@obolensk.org

**Keywords:** *Yersinia pestis*, outer membrane vesicles, protective antigen, plague vaccine

## Abstract

Despite the relatively low incidence of plague, its etiological agent, *Yersinia pestis*, is an exceptional epidemic danger due to the high infectivity and mortality of this infectious disease. Reports on the isolation of drug-resistant *Y. pestis* strains indicate the advisability of using asymmetric responses, such as phage therapy and vaccine prophylaxis in the fight against this problem. The current relatively effective live plague vaccine is not approved for use in most countries because of its ability to cause heavy local and system reactions and even a generalized infectious process in people with a repressed immune status or metabolic disorders, as well as lethal infection in some species of nonhuman primates. Therefore, developing alternative vaccines is of high priority and importance. However, until now, work on the development of plague vaccines has mainly focused on screening for the potential immunogens. Several investigators have identified the protective potency of bacterial outer membrane vesicles (OMVs) as a promising basis for bacterial vaccine candidates. This review is aimed at presenting these candidates of plague vaccine and the results of their analysis in animal models.

## 1. Introduction

The justification of antigenic composition of any modern vaccine is a primary and crucial step of its development. With regard to construction of such a medication for plague immunoprophylaxis, the problem of identifying new immunodominant antigens, and in some cases even epitopes that induce the formation of protective immunity, is extremely complex and has not been fully resolved as of yet. There are several reasons for this, and perhaps the most important one is that not all genes whose products are necessary for *Y. pestis* adaptation to different host niches during infection were able to be expressed in vitro to produce potential protective antigens in quantities sufficient for analysis [1]. In addition, in the process of antigen isolation and purification, as well as in the process of preparation of killed vaccines, the antigens’ native structure may change [2], causing conformational inactivation of their protective activity. It is also likely that, in addition to the already known immunodominant antigens, a large number of other antigens are involved in the induction of protective immunity to plague. Each of these antigens contributes relatively little to protective immunity (and is therefore difficult to be detected), but when taken as a whole group can provide sufficient protection.

As for the two immunodominant antigens of *Y. pestis*, subunit vaccines based on F1 (Caf1), the fraction 1 pilus antigen, and V (LcrV), a protein at the tip of type III secretion needles, have shown promising results in several animal models [3,4,5]. However, they mainly induce an immune response, which, as a rule, in the majority of animal models is significantly inferior in protective potency and duration of post-infectious immunity (in surviving animals) and even post-vaccination immunity induced by live attenuated bacteria in different mammalian species [6,7,8,9,10]. Only in mice subunit preparations containing F1 antigen can induce a degree of protection no less than that induced by the live plague vaccine. Conflicting data from several laboratories indicate that judging the level of animal protection by the titer of antibodies to a particular *Y. pestis* antigen should be done with great caution. There is both information that completely denies such a correlation [11] and information that shows a moderate or pronounced correlation [12,13]. These drawbacks of plague subunit vaccines explain why live plague vaccine prepared from derivatives of the *pgm*-negative *Y. pestis* strain EV76 continues to be used in several countries [14], despite concerns about the safety of live bacterial vaccines [15] and the widely reported lethal case of human plague caused by a similar *pgm*-negative strain in individuals with hemochromatosis [16]. One of the main advantages of subunit vaccines is the inclusion in their composition of only one or two immunodominant antigens. This low-component composition simplifies and reduces the cost of production and control of the remedy, but, in some cases, it can also be a significant drawback. For example, immunization with the capsular antigen F1 can in no way protect against infection with a non-capsulated strain that has retained virulence at the level of wild-type bacteria [17,18]. Both the primary antigens, F1 [19] and LcrV [20], have isoforms characterized by reduced cross-protection [19,21,22]. It is also necessary to consider the individual response of the immunized or infected animals and humans to the introduced antigens [4,23,24], as well as the specificity of the protective potency of individual *Y. pestis* antigens or their various combinations both in the composition of subunit [25,26,27,28,29,30,31,32,33,34,35] and live recombinant [36] vaccines with respect to the different mammalian species. The differences uncovered in the results of experiments on determining the protective efficacy of vaccine preparations with respect to monkeys are also likely to be due to using different species of these animals in the experiments: Rhesus monkeys (*Macaca mulatta*), hamadryads baboon (*Papio hamadryas*), vervet monkeys (*Cercopithecus pygerythrus*), or green monkeys (*Chlorocebus aethiops*); various mice lines also differ in sensitivity to *Y. pestis* and properties of immunogenesis [37].

At present, adjuvants are administered to increase the protective efficacy of vaccine preparations by generating a more balanced cellular and humoral immune response [38,39,40,41]. A number of natural bacterial cell components have been shown to have adjuvant activity [42,43,44,45,46,47]. Recently, a new approach for designing non-living bacterial vaccines based on the outer membrane vesicles (OMVs) technology was developed [48].

This review will examine the current state of yersiniae OMV vaccine research and discuss its potential application as a novel complex agent for conferring anti-infective protection.

## 2. Protective Potency of *Y. pestis* Antigens

The study of immunobiological (including protective) properties of individual components of *Y. pestis* cells began in the early 20^th^ century, shortly after the discovery of the plague pathogen. In these studies, the microbial biomass of *Y. pestis* was fractionated by several methods, and the protective potency of the obtained fractions against experimental plague was evaluated in experiments on laboratory animals. However, the beginning of the purposeful efforts to identify protective antigens of this microbe was laid by laboratories of H. Schutze [49,50] and K.F. Meyer [51,52]. The second group of researchers was able to isolate, purify, and characterize the physicochemical and immunobiological properties of fraction 1 (or F1 capsular antigen), the first of the identified “individual antigens” of *Y. pestis*. Fraction 1 (subfractions A and B) has been shown to be highly protective when used for active immunization of mice, but not guinea pigs [52]. Furthermore, a moderately efficient protection of the rhesus macaque infected with *Y. pestis* was observed after repeated immunizations with F1 capsular antigen in high, milligram doses [53], and production of complementary antibodies was induced by repeated immunization of volunteers with fraction 1 preparations [54]. The possibility of the formation of specific immunity to plague in guinea pigs is associated mainly with still unidentified water-insoluble components of the microbe [52,55].

The development and introduction into research practice of new physicochemical, molecular-genetic, and immunochemical methods have significantly expanded the possibilities of studying the mechanisms of immunogenesis, including identification of protective antigens. The spectrum of such *Y. pestis* antigens has increased significantly over the past decades. Table 1 presents a list of some of the native antigens providing, to one degree or another, protection of experimental animals infected with virulent strains of the plague pathogen.

It should be taken into account when assessing the data presented in Table 1 that in different studies, antigen preparations were obtained using various methodological approaches from cultures of several strains of *Yersinia* spp. grown under different conditions. Variant immunization schemes (multiplicity, dose, method of administration of antigens), conditions of challenge (method and time of infection, test-infecting strain, and dose of the pathogen), laboratory animal species, lines, conditions of their keeping, methods for assessing protection (number of surviving animals, life span to death), and statistical processing techniques of the results were used.

In this regard, it does not seem entirely correct to give a reliable comparative assessment of the immunogenicity of these antigens, especially in relation to their possible use in the composition of novel plague subunit vaccines. Therefore, the protective potency of the antigens in Table 1 is expressed qualitatively: “+”—antigen is protective; “−”—non-protective; “?”—no data. It should be emphasized that Table 1 includes only information on the active immunization with antigens isolated from *Y. pestis* or recombinant *Escherichia coli*.

Using other methodological approaches, the significance of a number of other proteins of the *Y. pestis* outer membrane in active and passive protection of rodents from plague was also shown. For example, passive immunization of mice with preparations of antibodies to Ail/OmpX and OmpA increased the survival rate of animals after subcutaneous infection, while anti-Pla protected mice from intranasal challenge. In all cases, the animals were infected with F1-capsule-negative *Y. pestis* strain [84]. It was shown that three-fold intranasal immunization with synthetic fragment of the YopE protein, YopE_69–77_, mixed with cholera toxin as an adjuvant, protected mice from primary pneumonic plague [85]. Immunization with the YopE_69–77_ fragment also protected mice from the infection with pseudotuberculosis [86]. Mice were protected from bubonic plague by immunization with the pesticin receptor Psn [87], while the combination of Psn with LcrV and F1 protected these animals from intranasal infection with *Y. pestis* [88].

As can be seen from the above literature data, protective properties have been shown for a rather large number of *Y. pestis* surface antigens. This unambiguously testifies to the multicomponent nature of the immunizing action of whole-cell plague vaccines. At the same time, there are publications in which the protectiveness of some of the aforementioned antigens is denied.

## 3. Animal Models for Testing Potency of Plague Vaccines

The above results were obtained mostly on animals of the same species—mice. The presence of such antigens, apparently, will not be sufficient for the formation of immunity in guinea pigs and primates, or many of them may not be protective at all. Currently, real attempts to design a subunit plague vaccine are associated with the use of only F1 and V antigens. The effectiveness of immunization with these two antigens was evaluated in experiments not only on mice, but also on animals of other species—rats, guinea pigs, non-human primates. As it turned out, the protective potency of antigens depended on the species of laboratory animals [52,53,89]. Several plague subunit vaccine candidates have been going through various stages of clinical trials for some years [90,91], but only one of them, the plague molecular microencapsulated vaccine, has now received a registration certificate in Russia [92]. After two subsequent vaccinations of volunteers, 67% of them developed specific antibody titers equal to or exceeding the threshold level, while 33% of the subjects did not develop specific antibodies to either V or F1 antigens. Seronegative individuals differed from responding ones by alleles of single nucleotide polymorphisms in 14 of immune response genes [4].

The choice of an adequate animal model is a key problem in research aimed at designing plague vaccines. For more than a century of research on the development of plague vaccines, based on individual fractions of the *Y. pestis* cell, animals of many species were used as model organisms. These were mice, rats, guinea pigs, rabbits, and primates, characterized not only by different susceptibility to the causative agent of plague, but also by different potency of the formed immunity. The vast majority of modern studies on the mechanisms of plague immunogenesis are carried out in mice. Proponents of this approach argue their position on the basis of the relative cheapness of animals of this species, the convenience of working with smaller animals, and less pronounced scatter in the results of determining the immunogenicity of antigens [93].

However, the results of such studies provide a one-sided view of the problems under consideration. In our opinion, at the initial stages of research of this kind, two species of animals should be used, mice and guinea pigs. This choice is determined by several circumstances. The main reason is that, with approximately equally high susceptibility to infection with the plague pathogen, the mechanisms of adaptive immunity to this infection in rodents of the named species differ sharply. While immunization with the live plague vaccine induces highly potent immunity in animals of the both species, antigens such as F1 and, to a lesser extent, V are protective for mice and less effective in immunizing guinea pigs [52,57]. In our opinion, the guinea pig is a more adequate model for studying the immunity to plague in primates and, likely, in humans, as compared to the mouse. This is indicated, in particular, by the low protectiveness of the F1 antigen during the primary immunization of guinea pigs and primates. The experience of developing and applicating plague vaccines in experiment and anti-epidemic practice clearly indicates that full-fledged human immunity to plague can be created only with the help of a medicine that includes several protective antigens. Furthermore, according to J. Keppie et al. [55,94] and other researchers, the ideal plague vaccine should include antigens that are protective for the both mice and guinea pigs as mandatory components.

Through the efforts of several groups of researchers [52,54,57], the significance of the *Y. pestis* water-soluble antigen F1 in resistance to plague in mice, and to a much lesser extent in guinea pigs, monkeys, and, seemingly, humans, has been established. Since the capsular antigen F1 is an excellent immunogen for mice infected with “classical” strains of *Y. pestis*, its inclusion in the composition of the vaccine being created does not cause doubts among the majority of researchers dealing with this problem. This choice is supported by experimental data on the high efficiency of F1 as a revaccination agent for guinea pigs [56] and hamadryas baboon (*Papio hamadryas*) [59], primarily immunized with the live plague vaccine. This can be important when developing long-term immunization regimens for various human populations. At the same time, it is not possible to achieve a pronounced protection of guinea pigs and primates after their primary immunization with this antigen [57,95]. In addition, it is obvious that the use of a monocomponent vaccine based on the F1 antigen cannot be effective in protecting animals infected with virulent F1-negative strains of the plague pathogen. Such strains are periodically isolated in natural plague foci [96,97] and even from people [98].

In our opinion, the presence of F1 and V antigens in the composition of acellular plague vaccines being developed is insufficient for the formation of highly potent immunity in guinea pigs, primates, and humans. Of the two species of small rodents used for the primary assessment of the level of immunity to plague, mice and guinea pigs, the latter models the mechanisms of immunogenesis in primates to a greater extent. This is suggested by experimental data indicating a relatively low protective effectiveness of F1 antigen in primary immunization of guinea pigs and primates (but not mice) and a high booster efficiency of this antigen for guinea pigs [56] and hamadryas baboons (*Papio hamadryas*) [59] primarily vaccinated with a live EV vaccine. In addition, as will be shown below, immunization of guinea pigs and primates with *Y. pseudotuberculosis* OMVs that do not contain F1 and V antigens induces pronounced protection against infection with virulent strains of the plague pathogen; these preparations are completely non-protective for mice. The foregoing determines further attempts to search for new protective antigens that would provide reliable protection of people from plague as part of acellular multicomponent vaccines.

## 4. Outer Membrane Vesicles (OMVs) and the Development of Bacterial Vaccines

The ability to form and excrete OMVs into the extracellular space is a widespread property of microorganisms, such as archaea, fungi, and bacteria, both gram-positive and gram-negative [99,100]. They are, as a rule, spherical or oval in shape, less often irregular, and their sizes can vary over a wide range, from 7–10 to 250–300 nm [101].

OMVs of gram-negative bacteria are separated from the surface of the bacterial cell as cell-body-budding structures, in which the content of the predominant periplasm is surrounded by an outer membrane [102]. Gram-negative outer membrane (OM) is linked to the adjacent peptidoglycan layer by proteins, such as Lpp and OmpA [103]. OMVs formation begins with the outward bulging of the OM at areas of the bacterial surface, where these OM-peptidoglycan bonds are disrupted. Further budding is provided due to the accumulation of periplasmic proteins in the detachment site. In addition, membrane budding is activated by the local accumulation of curvature-inducing OM proteins in weakened areas [104]. Deletion of genes coding for proteins that form OM-peptidoglycan or OM-peptidoglycan-internal membrane bonds affects the size of the OMVs and the rate of vesiculation [105]. The chemical composition of OMVs can be very diverse. In addition to obligate membrane components (lipopolysaccharides (LPSs), phospholipids, peptidoglycans, membrane-bound proteins), they may contain periplasmic proteins, nucleic acids (DNA, RNA), ionic metabolites, and various signaling molecules [106]. In addition, they can include cytoplasmic proteins, as well as proteins associated with the cytoplasmic membrane [107]. Bacterial cells grown in liquid and solid nutrient media have the ability to form vesicles. As a rule, vesicles are isolated, from the supernatant liquid after low-speed centrifugation of submerged cultures, followed by microfiltration, concentration by ultrafiltration, precipitation, or purification by ultracentrifugation in a density gradient of cesium chloride (sucrose) or gel filtration [102].

The ability to form vesicles is extremely important in the physiology of bacteria. The list of functions performed by OMVs is wide and affects such aspects of the life support of the microbe as: participation in the interaction of bacterial cells (quorum sensing, horizontal gene transfer, bactericidal effect), protection from external stress factors (bacteriophages, antibiotics, etc.), the manifestation of pathogenetic properties against eukaryotic macroorganisms (virulence factors delivery to host cells, biofilm formation, adsorption of antimicrobial peptides, function as decoys which bind and remove antibodies and other bactericidal components in e.g., serum). OMVs also perform other important physiological functions, such as nutrient acquisition and export of unnecessary compounds [104,108,109].

Some properties of vesicles determine the broad prospects for their application in practical medicine. Of particular interest is the possibility of using OMV for drug delivery, as well as for the development of bacterial vaccines [110]. The latter line of research is based on such specific features of OMVs as “their immunogenic properties, self-adjuvanticity, and capacity for enhancement by recombinant engineering” [111]. Some extremely important advantages of such remedies are the native conformation of protective antigens within the vesicles that provides potent immunity, as well as the possibility of designing medicines based on the heterologous antigens, which significantly expands the scope of their intended use [112,113].

OMVs experimental vaccines have been developed and tested on laboratory rodents against a number of bacterial pathogens. More than 30 million people have been immunized with OMVs *Neisseria meningitidis* type B vaccine. Despite the general success, the use of such vaccines, however, has another significant limitation related to fact that they are mainly effective against the homologous strains of the pathogen [114].

## 5. OMVs-Based Vaccines against *Y. pestis*

Initially, frequently observed electron-microscopically-recorded small structures in bacterial preparations were regarded as artifacts, some kind of cellular debris. The first works devoted to the bacterial vesicles began to be published in the mid-60s of the last century [115,116]. After confirming OMVs existence in *Y. pestis* [117], the questions about the conditions for their formation and the possibility of exploiting them as a basis for plague vaccines arose. It was shown that the presence of a pigmentation (*pgm*) locus in the chromosome is not a significant factor for the processes of vesicle formation, while physiological levels of glucose induce membrane vesicle secretion and affect the lipid and protein composition of *Yersinia pestis* cell surfaces [117]. Δ*pgm* EV vaccine strain of *Y. pestis* formed OMVs when grown in both liquid and solid nutrient media under favorable “physiological” conditions (Figure 1) [118].

For many bacteria, the ability to produce vesicles is a constitutive quality that increases under adverse external influences [104]. Stress effects on *Y. pestis* cultures, namely, joint incubation with a specific Pokrovskaya’s plague bacteriophage and, to a lesser extent, gentamicin, effective in the treatment of plague, induce an increase in the proportion of cells producing vesicles [118].

The ability of *Y. pestis* to produce OMVs has not been used for the development of anti-plague vaccine until recently, although such a possibility was proposed [117]. However, it was suggested to use the vesicles produced by *Bacteroides thetaiotaomicron*, one of the bacterial commensals of the human intestine, as a heterogeneous carrier of the protective *Y. pestis* antigens F1 and V [119]. More recently, the first results of studies were published, which showed the possibility of using vesicles produced by *Y. pestis* cells for active immunization of mice against plague [120]. The authors of this work used a *Y. pestis* engineered strain that produces a detoxified 1-dephosphorylated hexa-acylated version instead of highly-inflammatory bisphosphoryl hexa-acylated lipid A (monophosphoryl lipid A, MPLA), and, as it turned out, has an increased ability to produce vesicles. On the basis of this non-reactogenic variant, they have constructed a strain characterized by a high content of V antigen in the excreted vesicles. Immunization with such vesicles induced not only a high level of antibodies to the V antigen, but also a pronounced protection of animals from subcutaneous, as well as intranasal infection with the virulent *Y. pestis* strain [120].

The second *Yersinia* species, for which the ability to form vesicles was established, is the causative agent of pseudotuberculosis. Over the course of preliminary studies on the choice of the strain producing the sought-for antigen(s) protective for guinea pigs, we evaluated the ability of both viable and inactivated bacteria of the closely related to *Y. pestis* species, *Yersinia pseudotuberculosis* and *Yersinia enterocolitica*, to induce protective immunity to plague. It turned out that the level of resistance to plague in guinea pigs immunized with viable cells of *Y. pseudotuberculosis* strain 164/84 turned out to be even higher than that achievable under similar conditions using *Y. pestis* vaccine strain EV. The ImD_50_ (immunizing dose protecting 50% of infected animals from death) values for the variant carrying the low-calcium-response plasmid pLcr (also termed pCD1, pCad, pVW, or pYV) [121] and the variant without this plasmid turned out to be approximately equal and did not exceed, on average, 100 CFU when guinea pigs were infected with *Y. pestis* at a sufficiently high dose of 1000–2000 LD_50_. Viable cells of the plasmid-free derivative of *Y. enterocolitica* were not immunogenic under comparable conditions. *Y. enterocolitica* E9, carrying its own pYVe or pCD1 plasmid from *Y. pestis* strain EV, provided guinea pigs some (albeit low) protection against plague. Moreover, low protection or its absence were registered for the *Y. pestis* EV derivatives, one of which carries only its own pCD1 plasmid and the other is a plasmid-free variant, respectively [122] (see Table 2 for more details).

Taking into account the aforementioned facts, the biomasses of two isogenic variants of *Y. pseudotuberculosis* 164/84, differing only in the presence or absence of the pLcr and produced by submerged cultivation, were used as sources of antigens protective for guinea pigs. The immunogenicity of preparations of two types was evaluated: (i) Sterilized by microfiltration of culture broths and (ii) bacterial cells obtained from the same culture broths, inactivated by the method used to prepare the killed USP vaccine. In general, it was found that during submerged cultivation of bacteria, the unidentified protective substance is able to be excreted into the nutrient medium, and to a much greater extent when using the culture of the plasmid-free variant, the maximum amount of this protective substance accumulated in the culture liquid by 36–48 h of cultivation at 37 °C. In these experiments, animals were immunized with preparations of culture fluid concentrates freed from microbial cells by low-speed centrifugation followed by microfiltration. On the contrary, the protectiveness of the inactivated cells of the plasmid-carrying variant, on average, was higher than that of the cells of the plasmid-free variant [73]. It has been suggested that pLcr can prevent the from-the-cell release of the antigens, which induce the development of protective immunity to plague.

In order to identify the cultivation conditions favorable for the biosynthesis of the *Y. pseudotuberculosis* antigens that are protective for guinea pigs, a number of experiments were carried out with cultivation of bacteria in a liquid nutrient medium, to a greater extent simulating the internal environment of the host organism. However, neither addition of intact guinea pig serum to the nutrient medium, nor cultivation of bacteria in a dialysis-membrane chamber implanted into the abdominal cavity of *Cavia porcellus*, nor the direct intraperitoneal cultivation of bacteria provided an increase in the protection per biomass unit of preparations in the form of inactivated cells or corresponding supernatants [73].

Rabbit hyperimmune polyclonal serum produced against *Y. pestis* was depleted by antigenic fractions lacking antigen(s) protecting guinea pigs from experimental plague. The resulting monoreceptor serum in the reaction of Ouchterlony [124] with yersiniae preparations protecting guinea pigs from *Y. pestis* infection gave only one precipitation line [73]. The antigen protective for guinea pigs was named with the letter “Б” of the Cyrillic alphabet (“B” in Latin transliteration). Using the methods of stepwise salting out, stepwise ultrafiltration, and stepwise isoelectric precipitation, it was shown that the B antigen is a fairly large-molecular antigen that includes high-molecular structures. It contains carbohydrate, lipid, and protein components, which are firmly linked to each other. An immunochemically detectable increase in the amount of B antigen excreted into the liquid nutrient medium is observed with a decrease in the cultivation temperature of the producer strain from 37 °C to 27 °C [125]. Only very recently, during the electron-microscopic study of B-antigen preparations, it was shown that they include large-sized formations, which, as it turned out, are OMVs [126]. Figure 2 shows a micrograph of a *Y. pseudotuberculosis* cell with an already budded vesicle.

It is likely that the aforementioned more pronounced protectiveness of culture fluid of the plasmid-free variant of *Y. pseudotuberculosis* as compared to the plasmid-carrying variant for guinea pigs against experimental plague can be explained by the presence of pLcr reducing the ability of bacteria to excrete vesicles containing protective antigens. *Y. pseudotuberculosis* cells can produce vesicles at the both surface and submerged cultivation. The level of vesicle formation, determined by the proportion of vesicle-producing cells, increased significantly when a suspension of the pseudotuberculosis diagnostic bacteriophage produced by Russian Research Anti-Plague Institute “Microbe” was added to the bacterial culture. It also turned out that pre-incubation of this bacteriophage with a suspension of vesicles isolated from the culture fluid of *Y. pseudotuberculosis* 474/1b led to a significant decrease in the plaque-forming activity of the phage against the cells of this strain. Such an effect was found for a vesicle preparation isolated from a culture that was grown at 10 °C, but not 37 °C [127]. Apparently, the noted temperature dependence may be explained by the differences in the chemical composition of the compared preparations of vesicles, first of all, by a significantly higher content of O-side chains on the surface of vesicles isolated from the “cold” culture of the pathogen.

At cultivation temperatures of 6–28 °C, *Y. pseudotuberculosis* produces the complete S-form of LPS, including the O-polysaccharide (O-antigen), consisting of repeating oligosaccharide units, and bound to the hydrophobic part, lipid A, through the core oligosaccharide. At 37 °C, bacteria produce mainly the R-form of LPS, which does not carry the O-antigen, and the SR-form, which contains only one unpolymerized O-polysaccharide unit bound to the core. Due to insertions or deletions in 5 of the 17 genes of the O-antigen cluster obtained from *Y. pseudotuberculosis* O: 1b, *Y. pestis* produces only the R-form of LPS, consisting of the core and lipid A. The R-form lipopolysaccharide structures in *Y. pestis* and *Y. pseudotuberculosis* strains grown at 20–28 °C were demonstrated to be identical and to undergo similar temperature-induced modifications, except for that at 37 °C *Y. pestis* is unable to incorporate the palmitoyl group and thus produces a lower acylated less immunostimulatory lipid A [128,129,130].

Evaluation of B antigen immunogenicity in laboratory animals showed that almost all the protectiveness of the culture fluid of *Y. pseudotuberculosis* is determined by the presence of the B antigen vesicles in it [73]. This means that the antigen protection is most likely associated with the component(s) of the outer membrane and/or periplasmic space, but not with the inner membrane and cytoplasmic content.

Single subcutaneous immunization of guinea pigs with B antigen, emulsified in incomplete Freund’s adjuvant, induced marked protection of animals against subcutaneous [72,73] and respiratory [72] challenge with *Y. pestis*, including its capsule-less variant. Thus, the index of resistance (the ratio of LD_50_ for immunized to LD_50_ for non-immunized animals) of guinea pigs subcutaneously infected with the culture of the capsule-forming *Y. pestis* strain was 808 [72]. The best protection of guinea pigs from primary pneumonic plague was provided by immunization of animals with a complex preparation, including antigens F1 and B (Table 3).

The results of experiments on non-human primates (*Papio hamadryas*) showed the significance of B antigen in induction of specific immunity to respiratory infection with *Y. pestis*. For example, a single subcutaneous injection of a complex of antigens B (340 μg by dry weight) and F1 (500 μg by protein) in incomplete Freund’s adjuvant protected 2 of 4 animals from respiratory challenge with virulent strain 1300 at a dose of 70–100 LD_50_. Two control monkeys, in identical conditions immunized with only F1 antigen at the same dose and with the same adjuvant, died from primary pneumonic plague [72]. In another experiment on primates of the same species, F1 antigen (500 µg) was used for immunization in a complex with B antigen taken in three different doses (340 µg, 170 µg, and 85 µg); the antigenic preparations, adsorbed on another, less effective adjuvant, aluminum hydroxide gel, were injected subcutaneously into the animals twice. As a result of the subsequent respiratory infection with *Y. pestis* in a dose of 200-300 LD_50_, half of the hamadryas baboons taken in the experiment survived (and did not look sick) [74].

The level of serum antibodies to *Y. pestis* antigens is not an absolute marker of the specific resistance of *Papio hamadryas* and guinea pigs to plague [74]. Apparently, when assessing the level of specific resistance to plague in an individual immunized with a plague vaccine, one should not overestimate the role of the antibody titer to known protective antigens of the pathogen. The dominant mechanism of specific protection of the mammalian organism from plague infection is cellular, not humoral immunity [131,132,133,134,135].

The B antigen was completely unprotective for mice infected with virulent strains of *Y. pestis* [72].

For a more detailed immunochemical characterization of B antigen, a panel of hybridomas producing monoclonal antibodies (MAbs) raised against *Y. pseudotuberculosis* surface antigens, including antigen B, was generated. As it turned out, among them were MAbs to the O-side chains of lipopolysaccharide (LPS) [129], as well as MAbs against different epitopes of membrane proteins of the pathogen [126]. The first of them reacted with whole bacterial cells, B antigen and LPS preparations. LPSs isolated from “cold” *Y. pseudotuberculosis* cultures, differing from lipooligosaccharides predominantly isolated from yersiniae grown at 37 °C by the presence of O-side chains in the full-length LPSs, were more efficiently detected. MAbs of the second group interacted with the whole bacterial cells and B antigen irrespective of the cultivation temperature, but not with LPSs. The similarity of the chemical composition of the B antigen and LPS in terms of carbohydrate, fatty acid compositions, as well as the totality of data characterizing the comparative immunochemical activity of the two antigenic preparations, B and LPS, showed that B antigen includes a lipopolysaccharide component. At the same time, a significant difference in the protein content (5–8% in antigen B versus ≤ 1.5% in LPS isolated by the water-phenolic method), determines the significant difference in the immunobiological properties of the two antigens.

This circumstance explains the pronounced protectiveness of the first of them in experiments on laboratory animals infected with *Y. pestis*, and the absence of such for LPS preparations. The inability of *Y. pestis* LPS to induce resistance to experimental plague should also be noted [136,137,138], which is not surprising since the chemical composition and structure of lipid A and the core of which practically does not differ from the analogous parts of the LPS molecule of *Y. pseudotuberculosis* [130,139]. Furthermore, *Y. pestis* LPS can induce an immunosuppressive effect, in particular, after revaccination of *P. hamadryas*, which were initially vaccinated with a live plague vaccine EV, and with a combined preparation of F1 and LPS, the protective effect turned out to be significantly lower compared to the use for revaccination of only F1 antigen in the same dose [59].

The totality of the above data allows the authors of the investigation to state that the protective effect of the vesicle preparation isolated from the culture fluid of *Y. pseudotuberculosis* (B antigen) is provided not by the components of the LPS included in their composition, but by the immunogenicity of the protein component of the vesicles [122]. Participation of LPS in immunogenesis cannot be ruled out as an antigen, but seems to act as an immunomodulator.

The revealed phenomenon of vesicle formation in *Y. pseudotuberculosis* [126] has been confirmed [140], and the protein composition and some biological properties of vesicles produced by bacteria of two *Y. pseudotuberculosis* strains were determined. It has been shown that the vesicles produced by YPIII carry a cytotoxic necrotizing factor on their surface, inducing actin rearrangement, multinucleation, and loss of viability of HeLa cells. Vesicles of the ATCC 29,833 strain, characterized by almost 70% homology of the protein composition with that of the YPIII strain, did not induce such an effect [140]. As is well known, the ability to form vesicles by bacterial cells or the intensity of this property depend on many factors, such as strain differences, cultivation conditions, etc. [100,141]. For these reasons, Kolodziejek et al. [117] managed to reveal this property in *Y. pestis*, but not in *Y. pseudotuberculosis* under the experimental conditions they used.

## 6. Directions for Future Research

The development of plague vaccines began immediately after the discovery of the causative agent of this infection in the late 19th century. During the third plague pandemic, the first killed and live vaccines saved millions of human lives. However, the first generation of vaccines had a number of drawbacks, the analysis of which made it possible to formulate the modern WHO requirements [142] for a new generation of plague vaccines. Here we present an overview of current research on one of the ways to develop a self-adjuvant vaccine based on *Yersinia* OMVs.

The demonstrated efficiency of immunization of mice [45], as well as guinea pigs and primates [72,74] with preparations of vesicles isolated from *Y. pestis* or *Y. pseudotuberculosis,* confirms the prospects of further research on the development of acellular plague vaccine. However, there are also objective difficulties in the way of designing vaccines based on OMVs, namely: the difficulty of achieving high levels of target protective antigens in the vesicles; relatively narrow specificity of protective antigens, determined by strain polymorphism of the pathogens; and the possible presence in OMVs preparations of immunosuppressive molecules and/or toxins that can weaken the required protective effect or cause side reactions, respectively [143]. There are also additional problems, such as the difficulties associated with insufficient knowledge of the processes of vesicle biogenesis, low standard of antigenic composition, size, and immunogenicity, and the relatively small yield of the final product per unit of bacterial biomass. Furthermore, severe toxicity, determined, in particular, by LPS lipid A, is a particular problem in the creation of vaccines based on native vesicles of gram-negative bacteria. The problem of reducing toxicity is recognized as the key in the development of improved live vaccines mainly for this reason [144,145]. Fortunately, there are reports on the construction of vesicle-producing strains with reduced toxicity and increased capacity for vesicle formation by genetically engineered modification of LPS [146,147]. An important aspect of the problem under discussion is the development of a vaccine characterized not only by high immunogenicity, but also by low reactogenicity. This is a common and significant problem in the development of live, inactivated, particulate, and subunit vaccines against most infections caused by the gram-negative bacteria. The most rational way to realize this possibility is to construct a strain-producer of vesicles with a detoxified form of LPS lipid A [45,145,148,149].

To reduce the likelihood of side reactions to the vaccine, harmful factors, such as plasminogen activator (Pla) and murine toxin, should be eliminated from the OMVs [45]. In silico analysis of *Y. pestis* proteins for potential to serve as allergens revealed 170 (5.22%) probable allergens of total 3256 ORFs of the vaccine strain *Y. pestis* EV NIIEG. Of these, 38 belong to the group of extracellular or outer membrane proteins [150]. It can be assumed that excluding these allergens from the OMVs composition may increase the hypoallergenicity of the vaccine. On the other hand, cleansing the genome of unnecessary reactogenic genes that may play an adjuvanting role could also reduce immunogenicity of the whole vaccine. Therefore, genome editing should be carried out in stages with the obligatory control of the results obtained to efficiently modulate the balance between reactogenicity and immunogenicity.

According to the opinion of a number of researchers dealing with the problems of the vaccine-based prevention of plague [82,151], it is not realistic to create a molecular or particle vaccine equal in effectivity to the live plague vaccine. It is likely that the lower intensity of immunity induced by subunit or particle vaccines will be quite sufficient to ensure the effectiveness of anti-epidemic measures, given the known main advantages of such treatments: (i) the possibility of combined use with antibiotics and (ii) a high speed of immunity formation. In this regard, our experience in creating a vaccine preparation based on *Y. pseudotuberculosis* vesicles indicates the advisability of using an adjuvant. Unfortunately, the aluminum hydroxide gel, a component of several vaccines approved for use in human [152], provided a moderate potentiating effect in comparison with the incomplete Freund’s adjuvant, which is not used in preparations intended for human vaccines. Therefore, one of the challenges in the development of such a medicine is the selection of an effective and low reactive adjuvant. One of the possible solutions to this problem may be the use of biodegradable carriers as an adjuvant. As an example, immunization of mice with a preparation of microspheres based on polylactide and/or polyglycolide, including *Y. pestis* protective antigens F1 and V, provided the formation of highly intense immunity due to the prolonged release of the active principle of the drugs after their single administration [153,154,155].

Flagellin of gram-negative bacteria, which is an agonist of the Toll-like receptor 5 (TLR5), is a potent inducer of innate immunity effectors that partake in development of adaptive immunity. In mice immunized intranasally or intratracheally with *Y. pestis* F1 antigen and recombinant *Salmonella* flagellin, there was a sharp increase in the anti-F1 plasma immunoglobulin G (IgG) titers as compared to the control mice immunized with the F1 antigen alone [43]. Evaluation of the efficiency of a fusion protein consisting of *Salmonella enterica* serovar Enteritidis FliC (flagellin), in which nucleotides 586 to 1134 were deleted, and *Y. pestis* antigens F1 and V, showed that the flagellin-F1-V fusion protein retains full activity, stimulating Toll-like receptor 5 in vitro, and also induces stable antigen-specific humoral immunity in mice and two monkey species [156]. The inclusion of yersiniae or *Salmonella* flagellin into OMVs in native form or as a part of fusion proteins would be a logical continuation of this cycle of studies.

In summary, we can highlight two main areas of further work on the design of self-adjuvant vaccine based on *Yersinia* OMVs:

(i) search for antigens (or even epitopes) providing protection not only for small laboratory animals but for humans;

(ii) inclusion in OMVs of adjuvants providing a maximal in force protective immune response.

## 7. Conclusions

The pronounced reactogenicity of live vaccines and the low immunogenicity of killed vaccines used in anti-epidemic practice to fight plague determine the search for other development tactics of new effective means of vaccine-based prevention of this disease in humans. One of the promising approaches in this direction is the design of subunit plague vaccines. However, despite more than half a century of related research, there is still no plague vaccine internationally approved for human immunization. The closest to this are preparations based on the F1 and V *Y. pestis* antigens, developed in several laboratories, which are at different stages of preclinical or clinical trials. In different animal species it is unlikely to achieve the level of protection characteristic of, for example, a live EV vaccine with such subunit treatments. Active primary immunization of mice and rats with a complex preparation, including the above antigens, forms a high level of protection of animals against infection with wild-type capsule-forming *Y. pestis* strains. At the same time, the protection for primates and guinea pigs, animals used to assess the immunogenicity of plague vaccines, is relatively low. In this regard, it seems important to conduct further research to find antigen (antigens) that efficiently protect these animal taxa from plague. The long history of such studies shows that identification and isolation of an antigen, specific and highly immunogenic for primates and guinea pigs, from the produced in vitro cell biomass in a sufficiently purified form is a difficult task.

It is likely that *Y. pestis* cells are not capable to produce the desired antigen in quantities sufficient for its detection and induction of pronounced immunity in inactivated vaccines or subcellular fractions under commonly used laboratory conditions of cultivation. The second reason may be related to the destruction or inactivation of epitopes that are significant for formation of resistance to plague in animals of these species, as a result of uncontrolled destruction of bacterial macromolecules and natural complexes of these biomolecules in the course of isolation and purification of such antigens. The low immunogenicity of killed vaccines can also be explained by the action of inactivating agents that violate the integrity and native state of the surface structures of the pathogen, which, is likely a necessary condition for development of a full-fledged immunization process. The high level of protective potency induced in experimental animals and humans by administration of the live plague vaccine, as well as post-infectious immunity, is explained by the cumulative effect of already known, and perhaps not yet identified, protective antigens integrated into the outer membrane of an intact viable microbial cell.

In this regard, in order to develop a non-living plague vaccine with a developer-controlled protective potency and low reactogenicity that would also be incapable of causing an infectious process in persons with weakened immunity or metabolic disorders, it seems relevant to use the extracellular yersiniae vesicles (OMVs) excreted by a living bacterial cell under standard, “physiological” in vitro conditions. This approach is also supported by the fact that to date, about 20 antigens have been identified that are capable, under certain conditions, to induce resistance to plague in experimental animals. Construction of a subunit vaccine that includes all these antigens, or even most of them, is unreasonable and practically unrealistic. It seems promising to design a vaccine preparation, whose main component is OMVs, which already include some of the protective antigens. Furthermore, it is possible to design producer strains, with OMVs that will include protective protein antigens either in their natural form or with an altered amino acid sequence that will provide the immune system with only protective epitopes, and exclude immunosuppressive or toxic domains from them. In gram-negative producers of OMVs, gene editing can be carried out in order to eliminate the endotoxicity of LPSs while maintaining their adjuvant properties.

This review presents literature data confirming the effectiveness of vaccine preparations based on OMVs, including those produced by yersiniae cells. As an example, the OMV preparation induces the immunity of guinea pigs to respiratory challenge with both the wild type and capsule-free *Y. pestis* strains. A single immunization of guinea pigs and hamadryads baboons with a complex preparation, including OMVs of *Y. pseudotuberculosis* and F1 antigen of *Y. pestis*, gives animals a pronounced resistance to respiratory challenge with a virulent strain of the plague pathogen in a high dose, ~800 and 70–300 LD_50_, respectively [73]. At the same time, on the way to the development of such a vaccine suitable for the immunization of humans, it is necessary to solve many complex interrelated problems, such as reducing the reactogenicity of the treatment, justifying its antigenic composition, adjuvant form, one person-dose, method and scheme of use, its place in the system of anti-epidemic measures, and development of production technology.

In general, despite the existing difficulties, the development of a plague vaccine based on OMVs is one of the priority directions for improving the means of specific prevention of plague.

## Figures and Tables

**Figure 1 biomolecules-10-01694-f001:**
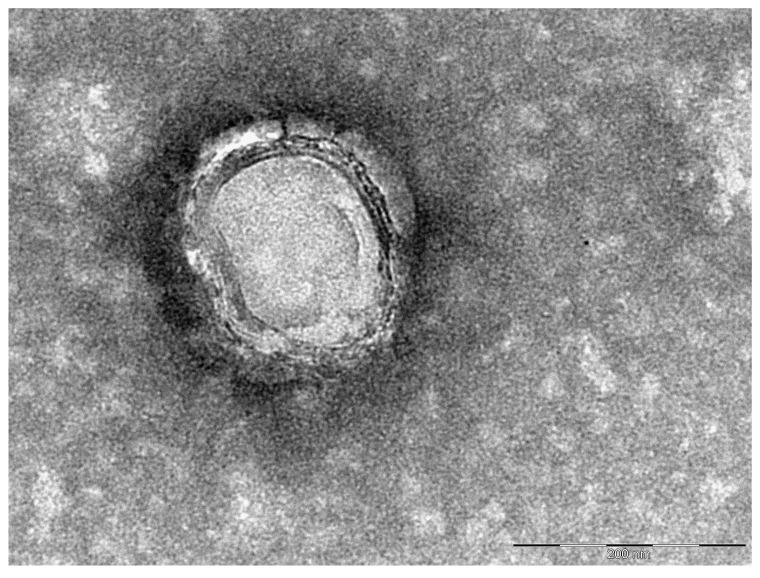
Transmission electron microscopy of separate vesicle produced by *Y. pestis* EV grown on solid nutrient medium.

**Figure 2 biomolecules-10-01694-f002:**
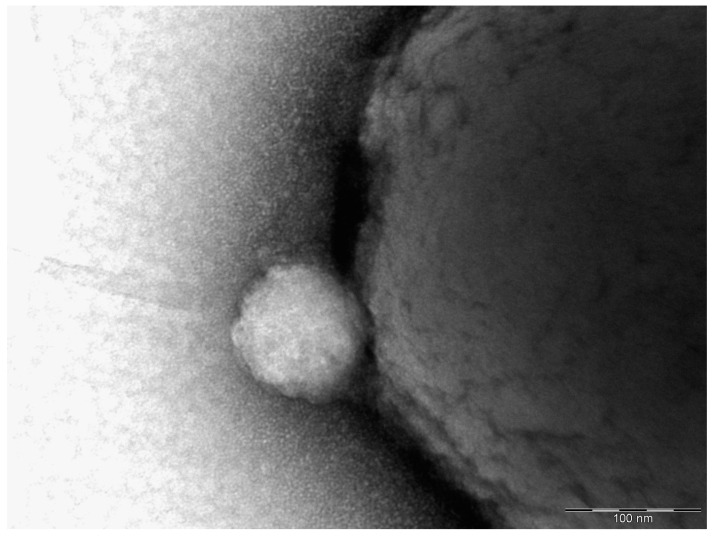
Vesicle budding from the cell body of *Y. pseudotuberculosis* 1b. Transmission electron microscopy, uranyl staining.

**Table 1 biomolecules-10-01694-t001:** Protective potency of *Y. pestis* native antigens for laboratory animals infected with the plague pathogen.

Antigen or Vaccine	Challenge Technique	Protective Potency for		
		Mice	Guinea Pigs	Nonhuman Primates
F1 (Caf1)	percutaneous ^1^	+ [52]	+ [52,56] ^2^	+ [57]
	respiratory	+ [58]	+ [27]	+ [53,59] ^2^
V (LcrV)	percutaneous	+ [60]	+ [28]	?
	respiratory	+ [60]	?	− [61] ^3^
«Murine» toxin	percutaneous	+ [62]	− [63]	?
	respiratory	?	?	?
Superoxide dismutase	percutaneous	?	+ [64]	?
	respiratory	?	?	?
YopD	percutaneous	+ [65]	?	?
	respiratory	?	?	?
YpkA	percutaneous	+ [65] ^4^	?	?
	respiratory	?	?	?
YscF	percutaneous	+ [66] ^5^	?	?
	respiratory	?	?	?
OppA	percutaneous	+ [67] ^4^	?	?
	respiratory	?	?	?
YadC	percutaneous	?	?	?
	respiratory	+ [68]	?	?
PsaA	percutaneous	− [69]	?	?
	respiratory	+ [70]	?	?
YP00606	percutaneous	+ [71]	?	?
	respiratory	?	?	?
YPO1914	percutaneous	+ [71]	?	?
	respiratory	?	?	?
YPO0612	percutaneous	+ [71]	?	?
	respiratory	?	?	?
YPO3119	percutaneous	+ [71]	?	?
	respiratory	?	?	?
YPO3047	percutaneous	+ [71]	?	?
	respiratory	?	?	?
YPO1377	percutaneous	+ [71]	?	?
	respiratory	?	?	?
YPCD1.05c	percutaneous	+ [71]	?	?
	respiratory	?	?	?
YPO0420	percutaneous	+ [71]	?	?
	respiratory	?	?	?
YPO3720	percutaneous	+ [71]	?	?
	respiratory	?	?	?
“B antigen” (OMVs)	percutaneous	− [72]	+ [72,73]	?
	respiratory	?	+ [72]	+ [72,74] ^6^
EV live vaccine ^7^	percutaneous	+ [75]	+ [76]	+ [77]
	respiratory	?	+ [78]	+ [79]
USP killed vaccine ^7^	percutaneous	+ [75]	+ [75]	+ [80]
	respiratory	− [81]	+ [27]	− [53]

^1^ Percutaneous mode of infection means the subcutaneous, intramuscular, or intradermal challenge. ^2^ Lebedinsky et al. [56] and Byvalov et al. [59] presented experimental evidence for the protective potency of the F1 antigen as a booster for animals primarily immunized with the EV live vaccine. ^3^ Li and Yang [61] gave indirect data on the V antigen low protection of primates based on the assessment of the effectiveness of their immunization with a F1+V complex preparation. ^4^ In these studies, the protective effect after inoculation with YpkA and OppA was recorded only by extending the life span to death in immunized animals after infection. ^5^ Protection of mice by intraperitoneal immunization with YscF is indicated for the intravenous route of infection. ^6^ The significance of the B antigen in immunogenesis is evidenced by the data on the protective potency of the complex of F1 and B antigens and lack of protection (under the conditions of this experiment) of the single F1 antigen administered at the same dose. ^7^ Prior plague vaccine attempts are described in [3,37,82,83]. “+”—antigen is protective; “−”—non-protective; “?”—no data.

**Table 2 biomolecules-10-01694-t002:** Influence of pLcr on the protective potency of viable *Yersinia* spp. strains for guinea pigs infected with *Y. pestis*.

Immunizing Strain ^1^	Presence of pLcr	Experiment Number	ImD_50_, CFU ^3^
*Y. enterocolitica* E9 (pYVe)	+ ^2^	1	17.1 × 10^6^
	+	2	11.5 × 10^6^
*Y. enterocolitica* E9 (pCD1)	+	1	25.3 × 10^6^
	+	2	9.9⋅× 10^6^
*Y. enterocolitica* E9 (plasmid-free)	-	1	>930 × 10^6^
*Y. pseudotubercul osis* 164/84 (pYV)	+	1	100
*Y. pseudotuberculosis* 164/84 (plasmid-free)	-	1	79
*Y. pestis* EV/1 (pCD1)	+	1	81.4 × 10^6^
*Y. pestis* EV/1 (plasmid-free)	-	1	>4300 × 10^6^

^1.^ The animals were immunized subcutaneously with yersiniae cultures. These strains were characterized earlier: *Y. enterocolitica* E9 and *Y. pestis* EV/1 in [123], *Y. pseudotuberculosis* 164/84 in [73]. ^2.^ “+”—presence, “−”—absence. ^3.^ Subcutaneous infection of guinea pigs with the plague pathogen (strain 1300 (LD_50_ for naïve guinea pigs is 12 ± 5 CFU)) at a dose of ~1000 LD_50_ was carried out three weeks after immunization.

**Table 3 biomolecules-10-01694-t003:** Protective potency of B and F1 antigens for guinea pigs respiratory challenged with wild-type or noncapsulated strains of *Y. pestis* [72].

Guinea Pigs Immunized with ^1^	*Y. pestis* Infecting Strain ^2^	Infecting Dose, CFU	Number of Survived/Infected Animals	Proportion of Survived Animals
B antigen	1300 (F1^+^) 67 (F1^−^)	1.54 × 10^5^ 1.04 × 10^5^	4/5 4/4	80 100
F1 antigen	1300 (F1^+^) 67 (F1^−^)	1.54 × 10^5^ 1.04 × 10^5^	0/6 0/6	0 0
B + F1 antigens	1300 (F1^+^) 67 (F1^−^)	1.54 × 10^5^ 1.04 × 10^5^	5/5 6/6	100 100
Naïve animals	1300 (F1^+^)	7 × 10^3^ 1.3 × 10^4^ 2.7 × 10^4^ 5.3 × 10^4^	2/5 1/5 0/5 0/5	40 20 0 0
	67 (F1^−^)	4.2 × 10^4^ 9.2 × 10^4^ 1.04 × 10^5^	1/4 0/3 0/5	25 0 0

^1.^ Animals were immunized once subcutaneously with antigens in Freud’s incomplete adjuvant. Doses of antigens per animal were: F1—100 μg by protein; B—34 μg by dry weight. ^2.^ Animals were challenged with *Y. pestis*: capsule-forming strain 1300 (F1^+^) and capsuleless strain 67 (F1^−^) [72].

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
