# Peer review of "Yersinia Outer Membrane Vesicles as Potential Vaccine Candidates in Protecting against Plague"

_biomolecules, 2020, doi:10.3390/biom10121694_

Round 1

Reviewer 1 Report

The authors of the review provide a review of many of the challenges associated with developing a vaccine for plague with a particular focus on recent research focused on the development of vaccines consisting of outer membrane vesicles from either Y. pestis or Y. pseudotuberculosis.  Overall, the content of the review is good.  The topic is interesting and worthy of publication.  I have no major concerns with the content; however, the text contains a lot of repetition that can be cut down with efficient writing.  Some sections, particularly discussion of the pros and cons of different animal models and prior vaccine work, are separated in many different areas. These can be reorganized into distinct sections, which will help cut down on repetition and shorten the article. 

Specific Comments below

Abstract line 31:  This review is aimed…

lines 45-48:  It is also likely that in addition to the already known immunodominant antigens, a large number of other antigens are involved in the induction of protective immunity to plague. “Each of these antigens contributes relatively little to protection and is therefore difficult to be detected.” 

Clarify the last sentence please.  Are you suggesting that a larger number of antigens contribute little individually to protective immunity, but when taken as a whole group provide sufficient protection?

ines 73-75:  It is not correct to classify cellular immunity as Th1 and humoral immunity as Th2. This incorrect and tired paradigm characterization has been disproven time and again.  There are cellular and humoral components to both Th1 and Th2 immunity.

line 128-29:  Does this mean none of the data presented includes studies that used recombinant antigens?  If not, clarify what is meant by ‘antigenic preparations isolated from cell biomass.’

145-146:  ‘Obviously, the presence of such antigens, apparently, will not be sufficient for the formation of..’

Why is this obvious?

190-210:  Downstream quality control for OMVs is also particularly difficult to maintain and is very important in the development of vaccines for clinical use. Standardization of OMV components may be difficult to achieve from batch to batch or lot to lot.

I suggest reorganizing some of the discourse on the use of different animal models into a separate section prior to discussing the OMV in plague.  I agree with the authors that it is very important to highlight, but it is stated in many different areas leading to unnecessary repetition and moving focus away from the discussion of OMVs in the plague section (4).

Also, the discussion of prior vaccine efficacy (such as the USP and F1 vaccines, again in section 4) can be consolidated into a separate section of the text. Possible section2, or another section entitled prior plague vaccine attempts.  Then in the remaining text it should be referred to.

Line 342-363:  This can be summarized more.  This is a review paper, t should be enough to briefly mention that the protective agent or B antigen was identified and was subsequently shown to be an OMV, and then reference.  Interested individuals can go back to the original source.

Line 387-419:  Text here needs to be cleaned up a bit, it jumps from pestis to pseudotuberculosis and guinea pig to mice to primate. Reorganize to make clearer.

Line 429, perhaps discuss the differential temperature regulation of Yersinia LPS, briefly as it is unsurprising that the LPS at 10C is different from that at 37C.  LPS produced by Y. pestis at 37C as well as pseudotuberculosis has no or very low reactivity

The authors use the word ‘obviously’ often.  In almost every instance, the observation being referred to would not be obvious to someone who does not have detailed knowledge of Yersinia bacteria and these usages should be removed.

484-489, some of the potentially reactogenic factors may play an adjuvanting role.  Removal of too many could also reduce immunogenicity of the whole vaccine.

494-497, it is unclear why vaccination in the presence of antibiotic treatment would be considered effective as an anti-epidemic measure.  Antibiotic use would not be proscribed unless a person were already infected, in which case vaccination is irrelevant.

497-498 adjuvants are not approved for use in humans (at least not by the FDA).  This is a common misconception.  Adjuvants are only approved as part of a vaccine formulation. Thus while alum is a component of many approved vaccine formulations, alum in of itself is not ‘approved’ for human use.  Regulatory agencies will consider any formulation based on safety and efficacy data.

475-516 The directions of future research section is actually a discussion of potential adjuvants, not really future research and should be renamed as such.

Author Response

Comments and Suggestions for Authors

The authors of the review provide a review of many of the challenges associated with developing a vaccine for plague with a particular focus on recent research focused on the development of vaccines consisting of outer membrane vesicles from either Y. pestis or Y. pseudotuberculosis. Overall, the content of the review is good.  The topic is interesting and worthy of publication.  I have no major concerns with the content; however, the text contains a lot of repetition that can be cut down with efficient writing.  Some sections, particularly discussion of the pros and cons of different animal models and prior vaccine work, are separated in many different areas. These can be reorganized into distinct sections, which will help cut down on repetition and shorten the article.

Response: Thank you for your careful reading of the manuscript and helpful comments.

Specific Comments below

Point 1: Abstract line 31:  This review is aimed…

Response 1: We have made a correction (line 34 of the revised version of our manuscript).

Point 2: lines 45-48:  It is also likely that in addition to the already known immunodominant antigens, a large number of other antigens are involved in the induction of protective immunity to plague. “Each of these antigens contributes relatively little to protection and is therefore difficult to be detected.”

Clarify the last sentence please. Are you suggesting that a larger number of antigens contribute little individually to protective immunity, but when taken as a whole group provide sufficient protection?

Response 2: We hope that now, with the help of the reviewer, we have managed to present our position to readers more clearly (lines 51-53 of the revised version of our manuscript).

Point 3: lines 73-75: It is not correct to classify cellular immunity as Th1 and humoral immunity as Th2. This incorrect and tired paradigm characterization has been disproven time and again.  There are cellular and humoral components to both Th1 and Th2 immunity.

Response 3: We have made a correction. “Th1” and “Th2” are excluded (line 87 of the revised version of our manuscript).

Point 4: line 128-129: Does this mean none of the data presented includes studies that used recombinant antigens?  If not, clarify what is meant by ‘antigenic preparations isolated from cell biomass.’

Response 4: We hope that the text has now been clarified (lines 166-168 of the revised version of our manuscript).

Point 5: 145-146: ‘Obviously, the presence of such antigens, apparently, will not be sufficient for the formation of..’

Why is this obvious?

Response 5: This is the cost of translation from Russian into English. Words ‘obviously’ and ‘obvious’ were removed from the text (line 186 of the revised version of our manuscript).

Point 6: 190-210:  Downstream quality control for OMVs is also particularly difficult to maintain and is very important in the development of vaccines for clinical use. Standardization of OMV components may be difficult to achieve from batch to batch or lot to lot.

Response 6: Yes, indeed, the problem of standardizing the component composition of OMVs at the stages of developing such a vaccine and, if successful, during its production is one of the key and difficult to solve in practice issues. To reinforce the importance of this problem, text lines 197-210 were moved to the section 5. Directions for future research (line 616-630 of the revised version of our manuscript).

Point 7: I suggest reorganizing some of the discourse on the use of different animal models into a separate section prior to discussing the OMV in plague.  I agree with the authors that it is very important to highlight, but it is stated in many different areas leading to unnecessary repetition and moving focus away from the discussion of OMVs in the plague section (4).

Response 7: In accordance with the recommendation of the reviewer, a separate section, Animal models for testing potency of plague vaccines was added to the manuscript (lines 185-252 of the revised version of our manuscript).

Point 8: Also, the discussion of prior vaccine efficacy (such as the USP and F1 vaccines, again in section 4) can be consolidated into a separate section of the text. Possible section2, or another section entitled prior plague vaccine attempts.  Then in the remaining text it should be referred to.

Response 8: Given the large length of the manuscript, we do not provide a detailed description of prior plague vaccine attempts, but refer the reader (Table 1, line 153 of the revised version of our manuscript) to recent reviews where this information is presented.

Point 9: Line 342-363:  This can be summarized more.  This is a review paper, it should be enough to briefly mention that the protective agent or B antigen was identified and was subsequently shown to be an OMV, and then reference.  Interested individuals can go back to the original source.

Response 9: To our regret, "interested individuals" will not be able to "go back to the original source" as lines 342-358 (lines 442-458 of the revised version of our manuscript) cite the publication of 2005, and publications from this Russian-language journal are available on the Internet only since 2008 (https://journal.microbe.ru/jour/issue/archive?issuesPage=2#issues).

Point 10: Line 387-419:  Text here needs to be cleaned up a bit, it jumps from pestis to pseudotuberculosis and guinea pig to mice to primate. Reorganize to make clearer.

Response 10: The text has been cleaned of redundant information regarding passive immunity and infection of immune animals with pseudotuberculosis (lines 498-524 of the revised version of our manuscript).

Point 11: Line 429, perhaps discuss the differential temperature regulation of Yersinia LPS, briefly as it is unsurprising that the LPS at 10C is different from that at 37C.  LPS produced by Y. pestis at 37C as well as pseudotuberculosis has no or very low reactivity

Response 11: The differential temperature regulation of the Yersinia LPS is now briefly discussed (lines 480-490 of the revised version of our manuscript).

Point 12: The authors use the word ‘obviously’ often.  In almost every instance, the observation being referred to would not be obvious to someone who does not have detailed knowledge of Yersinia bacteria and these usages should be removed.

Response 12: Words ‘obviously’ and ‘obvious’ were removed from the text.

Point 13: 484-489, some of the potentially reactogenic factors may play an adjuvanting role.  Removal of too many could also reduce immunogenicity of the whole vaccine.

Response 13: Thank you for your comment. Now this issue is briefly discussed in the manuscript (lines 642-646 of the revised version of our manuscript)..

Point 14: 494-497, it is unclear why vaccination in the presence of antibiotic treatment would be considered effective as an anti-epidemic measure.  Antibiotic use would not be proscribed unless a person were already infected, in which case vaccination is irrelevant.

Response 14: Antibiotic prophylaxis indicated for asymptomatic individuals in close contact (<2m), exposure to bodily fluids, or aerosolized deliberate release [Ansari I, Grier G, Byers M. Deliberate release: Plague - A review. J Biosaf Biosecur. 2020 Mar;2(1):10-22. doi: 10.1016/j.jobb.2020.02.001].

“Combining EV76-immunization and second-line antibiotic treatment, which are individually insufficient, led to a synergistic protective effect that represents a proof of concept for efficient combinational therapy in cases of infection with antibiotic-resistant strains” [Vagima Y, Tidhar A, Chitlaru T, Mamroud E. Postexposure Administration of a Yersinia pestis Live Vaccine for Potentiation of Second-Line Antibiotic Treatment Against Pneumonic Plague. J Infect Dis. 2019 Aug 30;220(7):1147-1151. doi: 10.1093/infdis/jiz260].

Point 15: 497-498 adjuvants are not approved for use in humans (at least not by the FDA).  This is a common misconception.  Adjuvants are only approved as part of a vaccine formulation. Thus while alum is a component of many approved vaccine formulations, alum in of itself is not ‘approved’ for human use.  Regulatory agencies will consider any formulation based on safety and efficacy data.

Response 15: Thank you for this comment. The text has been corrected (lines 654-655 of the revised version of our manuscript). We will take this into account for the future.

Point 16: 475-516 The directions of future research section is actually a discussion of potential adjuvants, not really future research and should be renamed as such.

Response 16: Perhaps the adjuvant properties of OMVs are their main function in the composition of vaccines. However, we have also added a brief description of the expected works on optimizing the antigenic composition of promising yersiniae OMVs vaccines aimed at expanding the range of effectively immunized mammalian species (namely, humans) (lines 680-732 of the revised version of our manuscript).

Reviewer 2 Report

The review article entitled, “Yersinia outer membrane vesicles as potential vaccine candidates in protecting against plague” is an attempt to describe the protective potency and possible application of bacterial outer membrane vesicles (OMVs) as promising vaccine candidates against Y. pestis infections. Whereas the current manuscript was unable to maintain this concept and mixed multiple unrelated concepts. The manuscript is presented predominantly with the author’s own hypothesis rather previous findings published by different research group and that actually derailed the entire story line of this review article. With the presence of large number of grammatical mistakes along with lengthy and complicated statements, it very hard for the reader to concentrate and understand the actual concept of manuscript. A professional language editing is need for the entire manuscript. Unfortunately, I cannot recommend publication of the manuscript in its current form. Please find below a detailed list of amendments requested for modification of the manuscript to the extent that it may be suitable for publication.

  1. Authors may need to briefly describe backgrounds and current progress of OMVs in your introduction part, which is in line with your title.
  2. lines 49-60, it is not clear what is the main point they want to state. It is so confused to this sentence: “One of the main advantages of subunit vaccines, the inclusion in their composition of only one or two immunodominant antigens, in some cases can be a significant drawback.”
  3. It is hard to connect the second part “2. Protective potency of Y. pestis antigens” with the Yersinia OMV vaccine development.
  4. The quality of Figure 1 is poor.
  5. Lines 295-319, hard to follow.
  6. lines 450-456, “… provided not by the components of the LPS included in their composition, but by the immunogenicity of the protein component of the vesicles…” It is not accurate. Do you try to remove LPS from OMVs using detergent? Authors may use “O-antigen” instead of “LPS”

  1. lines 491-493, “…. it is not realistic to create a molecular or particle vaccine equal in its effectivity to the live plague vaccine…” If it is not realistic, why authors claimed “OMVs is one of the priority directions” at the end of this manuscript. Actually, in Wang et al (2020) paper, recombinant Y. pestis OMVs enclosing considerable amounts of LcrV could induced robust immune responses and protection than subunit vaccine or even better than the Pgm- strain presented in their previous studies.

  1. References are used inappropriately i.e., Ref#3 used for the application of F1 and LcrV in different animal model. Reference 4,5 for short term and humoral concerted immune responses against F1 and LcrV and many more. Repetitive usage of same reference (i.e., 110 and 38; 26 and 80) are common. Several recent findings on the topic are skipped and around 30% of reference that are older than 15-20 years are incorporated.

Author Response

Point 1: The review article entitled, “Yersinia outer membrane vesicles as potential vaccine candidates in protecting against plague” is an attempt to describe the protective potency and possible application of bacterial outer membrane vesicles (OMVs) as promising vaccine candidates against Y. pestis infections. Whereas the current manuscript was unable to maintain this concept and mixed multiple unrelated concepts. The manuscript is presented predominantly with the author’s own hypothesis rather previous findings published by different research group and that actually derailed the entire story line of this review article. With the presence of large number of grammatical mistakes along with lengthy and complicated statements, it very hard for the reader to concentrate and understand the actual concept of manuscript. A professional language editing is need for the entire manuscript. Unfortunately, I cannot recommend publication of the manuscript in its current form. Please find below a detailed list of amendments requested for modification of the manuscript to the extent that it may be suitable for publication.

Response 1: Thank you for your careful reading of the manuscript and helpful comments. We hope that after making the edits recommended by the reviewer, the manuscript became suitable for publication.

Point 2: Authors may need to briefly describe backgrounds and current progress of OMVs in your introduction part, which is in line with your title.

Response 2: According to most reviewers, our review is a bit long. We tried to reduce it as much as possible. The background and current progress of OMVs is presented in part 4 of the manuscript. Even their brief duplication in our introductory part will lead to an unjustified expansion of the text.

Point 3: lines 49-60, it is not clear what is the main point they want to state. It is so confused to this sentence: “One of the main advantages of subunit vaccines, the inclusion in their composition of only one or two immunodominant antigens, in some cases can be a significant drawback.”

Response 3: We have added text indicating the benefits of low-component subunit vaccines. As for their shortcomings, there are 12 lines (lines 72-85) dedicated to them and following the sentence that caught your attention.

Point 4: It is hard to connect the second part “2. Protective potency of Y. pestis antigens” with the Yersinia OMV vaccine development.

Response 4: Taking into account the possibility of including into the composition of OMVs several protective Y. pestis antigens we consider in section 2 the most promising of them.

Point 5: The quality of Figure 1 is poor.

Response 5: Our photos

are of the same quality as OMVs micrographs from publications:

  • Toyofuku, M., Nomura, N. & Eberl, L. Types and origins of bacterial membrane vesicles. Nat Rev Microbiol 17, 13–24 (2019). https://doi.org/10.1038/s41579-018-0112-2
  • Microbiol., 26 June 2020 | https://doi.org/10.3389/fmicb.2020.01268

Point 6: Lines 295-319, hard to follow.

Response 6: To shorten and simplify the text, we excluded the paragraph (lines 295-304), as not directly related to the problem under discussion.

Point 7: lines 450-456, “… provided not by the components of the LPS included in their composition, but by the immunogenicity of the protein component of the vesicles…” It is not accurate.

Response 7: Given that a review article is offered to the readers' attention, we cannot be held responsible for the opinions of the authors of the cited publications. We can only cite them and express our (your) point of view on this matter. :)

Do you try to remove LPS from OMVs using detergent? Authors may use “O-antigen” instead of “LPS”

It seems to us that removing LPS from OMVs using a detergent will lead to the destruction of the outer membrane, the outer layer of which is largely composed of LPS, and, accordingly, to the destruction of the vesicles themselves.

Point 8: lines 491-493, “…. it is not realistic to create a molecular or particle vaccine equal in its effectivity to the live plague vaccine…” If it is not realistic, why authors claimed “OMVs is one of the priority directions” at the end of this manuscript. Actually, in Wang et al (2020) paper, recombinant Y. pestis OMVs enclosing considerable amounts of LcrV could induced robust immune responses and protection than subunit vaccine or even better than the Pgm- strain presented in their previous studies.

Response 8: In justifying your comment, you omitted the first part of the sentence, "According to the opinion of a number of researchers dealing with the problems of the vaccine-based prevention of plague [81, 142], it is not realistic to ...", which explicitly states that this is the opinion of only a part of the researchers. We agree with the opinions of the second part of the plague hunters.

Point 9: References are used inappropriately i.e., Ref#3 used for the application of F1 and LcrV in different animal model. Reference 4,5 for short term and humoral concerted immune responses against F1 and LcrV and many more.

Response 9: We apologize for technical errors in citations, we have made the appropriate corrections.

Repetitive usage of same reference (i.e., 110 and 38; 26 and 80) are common.

It is not clear to us how the negative attitude of the reviewer to repetitive usage of same reference can be explained. This is a fairly common occurrence in various scientific journals: (i) ref. 22 in Tourlomousis, P., Wright, J.A., Bittante, A.S. et al. Modifying bacterial flagellin to evade Nod-like Receptor CARD 4 recognition enhances protective immunity against Salmonella. Nat Microbiol 5, 1588–1597 (2020). https://doi.org/10.1038/s41564-020-00801-y, (ii) ref. 48 in Sable SB, Posey JE, Scriba TJ. Tuberculosis Vaccine Development: Progress in Clinical Evaluation. Clin Microbiol Rev. 2019 Oct 30;33(1):e00100-19. doi: 10.1128/CMR.00100-19. PMID: 31666281, etc.

Several recent findings on the topic are skipped and … around 30% of reference that are older than 15-20 years are incorporated.

We did not find in the available literature new publications on the properties and applications of OMVs of Yersinia. At present, almost all candidate plague vaccines include native F1 and V antigens, a fusion protein based on them, their protective epitopes or genes encoding them. In a purified form, these immunodominant antigens were isolated and characterized in the middle of the last century, which is significantly more than 15-20 years.

Point 1: The review article entitled, “Yersinia outer membrane vesicles as potential vaccine candidates in protecting against plague” is an attempt to describe the protective potency and possible application of bacterial outer membrane vesicles (OMVs) as promising vaccine candidates against Y. pestis infections. Whereas the current manuscript was unable to maintain this concept and mixed multiple unrelated concepts. The manuscript is presented predominantly with the author’s own hypothesis rather previous findings published by different research group and that actually derailed the entire story line of this review article. With the presence of large number of grammatical mistakes along with lengthy and complicated statements, it very hard for the reader to concentrate and understand the actual concept of manuscript. A professional language editing is need for the entire manuscript. Unfortunately, I cannot recommend publication of the manuscript in its current form. Please find below a detailed list of amendments requested for modification of the manuscript to the extent that it may be suitable for publication.

Response 1: Thank you for your careful reading of the manuscript and helpful comments. We hope that after making the edits recommended by the reviewer, the manuscript became suitable for publication.

Point 2: Authors may need to briefly describe backgrounds and current progress of OMVs in your introduction part, which is in line with your title.

Response 2: According to most reviewers, our review is a bit long. We tried to reduce it as much as possible. The background and current progress of OMVs is presented in part 4 of the manuscript. Even their brief duplication in our introductory part will lead to an unjustified expansion of the text.

Point 3: lines 49-60, it is not clear what is the main point they want to state. It is so confused to this sentence: “One of the main advantages of subunit vaccines, the inclusion in their composition of only one or two immunodominant antigens, in some cases can be a significant drawback.”

Response 3: We have added text indicating the benefits of low-component subunit vaccines. As for their shortcomings, there are 12 lines (lines 72-85) dedicated to them and following the sentence that caught your attention.

Point 4: It is hard to connect the second part “2. Protective potency of Y. pestis antigens” with the Yersinia OMV vaccine development.

Response 4: Taking into account the possibility of including into the composition of OMVs several protective Y. pestis antigens we consider in section 2 the most promising of them.

Point 5: The quality of Figure 1 is poor.

Response 5: Our photos

are of the same quality as OMVs micrographs from publications:

  • Toyofuku, M., Nomura, N. & Eberl, L. Types and origins of bacterial membrane vesicles. Nat Rev Microbiol 17, 13–24 (2019). https://doi.org/10.1038/s41579-018-0112-2
  • Microbiol., 26 June 2020 | https://doi.org/10.3389/fmicb.2020.01268

Point 6: Lines 295-319, hard to follow.

Response 6: To shorten and simplify the text, we excluded the paragraph (lines 295-304), as not directly related to the problem under discussion.

Point 7: lines 450-456, “… provided not by the components of the LPS included in their composition, but by the immunogenicity of the protein component of the vesicles…” It is not accurate.

Response 7: Given that a review article is offered to the readers' attention, we cannot be held responsible for the opinions of the authors of the cited publications. We can only cite them and express our (your) point of view on this matter. :)

Do you try to remove LPS from OMVs using detergent? Authors may use “O-antigen” instead of “LPS”

It seems to us that removing LPS from OMVs using a detergent will lead to the destruction of the outer membrane, the outer layer of which is largely composed of LPS, and, accordingly, to the destruction of the vesicles themselves.

Point 8: lines 491-493, “…. it is not realistic to create a molecular or particle vaccine equal in its effectivity to the live plague vaccine…” If it is not realistic, why authors claimed “OMVs is one of the priority directions” at the end of this manuscript. Actually, in Wang et al (2020) paper, recombinant Y. pestis OMVs enclosing considerable amounts of LcrV could induced robust immune responses and protection than subunit vaccine or even better than the Pgm- strain presented in their previous studies.

Response 8: In justifying your comment, you omitted the first part of the sentence, "According to the opinion of a number of researchers dealing with the problems of the vaccine-based prevention of plague [81, 142], it is not realistic to ...", which explicitly states that this is the opinion of only a part of the researchers. We agree with the opinions of the second part of the plague hunters.

Point 9: References are used inappropriately i.e., Ref#3 used for the application of F1 and LcrV in different animal model. Reference 4,5 for short term and humoral concerted immune responses against F1 and LcrV and many more.

Response 9: We apologize for technical errors in citations, we have made the appropriate corrections.

Repetitive usage of same reference (i.e., 110 and 38; 26 and 80) are common.

It is not clear to us how the negative attitude of the reviewer to repetitive usage of same reference can be explained. This is a fairly common occurrence in various scientific journals: (i) ref. 22 in Tourlomousis, P., Wright, J.A., Bittante, A.S. et al. Modifying bacterial flagellin to evade Nod-like Receptor CARD 4 recognition enhances protective immunity against Salmonella. Nat Microbiol 5, 1588–1597 (2020). https://doi.org/10.1038/s41564-020-00801-y, (ii) ref. 48 in Sable SB, Posey JE, Scriba TJ. Tuberculosis Vaccine Development: Progress in Clinical Evaluation. Clin Microbiol Rev. 2019 Oct 30;33(1):e00100-19. doi: 10.1128/CMR.00100-19. PMID: 31666281, etc.

Several recent findings on the topic are skipped and … around 30% of reference that are older than 15-20 years are incorporated.

We did not find in the available literature new publications on the properties and applications of OMVs of Yersinia. At present, almost all candidate plague vaccines include native F1 and V antigens, a fusion protein based on them, their protective epitopes or genes encoding them. In a purified form, these immunodominant antigens were isolated and characterized in the middle of the last century, which is significantly more than 15-20 years.

Reviewer 3 Report

This is  a comprehensive review, but rather long. I suggest this review could be considerably shortened without loss of impact.

My major question about the OMV approach is: how do the authors propose to make it scalable? could it ever become a recombinant process? The difficulty with such an approach to a vaccine is the batch-to-batch variation plus the laborious, potentially hazardous microbiological processes involved.

Additional questions/comments:

abstract line 25: 'capability' should be 'ability'

Introduction lines 51-54: 'sub-unit vaccines mainly induce a short-lived humoral response': this depends on the formulation and excipients, a fact which should be mentioned. see Williamson et al Infect Immun 2005, 73, 3598-3608 for human immune response to sub-unit vaccine out to 3 months with evidence of functional antibody at this timepoint; also Williamson et al 2007 which showed that functional antibody was maintained in macaques for >1year, without further boosting ( Microb Pathogenesis 42, 12-22) .

Line 62: 'Both the....'

line 68: 'protectivity' should be 'protective efficacy'

Line 128 and Table 1: these are all native  proteins: tat should be made clear in the table title

Lines 151-152 In my own experience guinea pigs are not a good model for plague. It may be unethical to suggest that more species are needed in vaccine efficacy testing -certainly species such as baboons. See also lines 273-4: essential to show efficacy in guinea pigs as well as mice?  how about an appropriate NHP species (e.g. macaque?)

Lines 334-341: what is the 'protective substance' referred to here from Y.pseudotuberculosis? These lines are difficult to follow....

Lines 419: Babbons...'did not feel sick'. How did these authors know (unless baboons can talk?). this is an anthropomorphic comment and should be removed.

Lines 420-425: no doubt cellular immunity is required (both humoral and cellular immunity are required for protection) but the cellular component is overstated here

Line 526: why is it 'obviously'? EV vaccines are efficacious but as mentioned previously in this review can cause severe side effects. Sub-unit vaccines(depending on formulation) can provide a safe and efficacious alternative and can be produced in large consistent quantity without hazard. Please can the authors present a more balanced argument. Then the case for OMV's can be assessed more effectively.

Author Response

Comments and Suggestions for Authors

Point 1: This is a comprehensive review, but rather long. I suggest this review could be considerably shortened without loss of impact.

Response 1: Thank you for your careful reading of the manuscript and helpful comments.

Yes, the manuscript is indeed quite long. This is due to the fact that it includes a review of not only readily available English-language sources, but also a number of Russian-language publications that are inaccessible to the vast majority of readers, but include data from rather expensive experiments using guinea pigs and non-human monkeys, as well as an alternative logic of experimental research aimed to develop an effective plague vaccine. We tried to shorten the text of the manuscript whenever possible, but the answers to the questions and comments of the reviewers practically canceled our efforts.

My major question about the OMV approach is: how do the authors propose to make it scalable?

The OMV approach is already scalable. Lines 308-311: More than 30 million people have been immunized with OMVs Neisseria meningitidis type B vaccine … [103].

could it ever become a recombinant process?

OMVs experimental recombinant vaccines against a number of bacterial pathogens have been developed and tested on laboratory rodents:

Antibody-mediated immunity induced by engineered Escherichia coli OMVs carrying heterologous antigens in their lumen

L Fantappiè, M De Santis, E Chiarot… - Journal of …, 2014 - Taylor & Francis

… As shown in Fig. 5, the 3 recombinant OMVs induced antigen-specific antibody responses which,
in the case of Slo dm -OMVs and SpyCEP-OMVs, were in the same range as the titers elicited
by 20 µg doses of the corresponding recombinant antigens formulated in Alum …

[PDF] tandfonline.comFull View

Recombinant outer membrane vesicles carrying Chlamydia muridarum HtrA induce antibodies that neutralize chlamydial infection in vitro

E Bartolini, E Ianni, E Frigimelica… - Journal of …, 2013 - Taylor & Francis

… To the best of the authors' knowledge, this is the first demonstration that, by virtue of their ability
to carry properly folded membrane-associated proteins, recombinant OMVs can be exploited
to induce functional immune responses that otherwise would be difficult to elicit …

[HTML] plos.orgFull View

[HTML] Mechanistic insight into the TH 1-biased immune response to recombinant subunit vaccines delivered by probiotic bacteria-derived outer membrane vesicles

JA Rosenthal, CJ Huang, AM Doody, T Leung… - PloS one, 2014 - journals.plos.org

… Noting that the untapped potential of recombinant OMVs aligned well with key limitations of subunit
vaccines, we investigated whether a T H 1-biased recombinant vaccine delivery platform could
be created from OMVs derived from Gram-negative probiotic bacteria …

Recombinant protein meningococcal serogroup B vaccine combined with outer membrane vesicles

X Bai, J Findlow, R Borrow - Expert opinion on biological therapy, 2011 - Taylor & Francis

… Several approaches have been taken to increase the breadth of coverage of OMV vaccines. One
has been the use of recombinant OMVs to express multiple PorA antigens and led to the
development of HexaMen and NonaMen by the Netherlands Vaccine Institute …

[PDF] pnas.orgFree from Publisher

Delivery of foreign antigens by engineered outer membrane vesicle vaccines

DJ Chen, N Osterrieder, SM Metzger… - Proceedings of the …, 2010 - National Acad Sciences

Skip to main content. Submit; About: Editorial Board; PNAS Staff; FAQ; Accessibility Statement;
Rights and Permissions; Site Map. Contact; Journal Club; Subscribe: Subscription Rates;
Subscriptions FAQ; Open Access; Recommend PNAS to Your Librarian. Log in; Log out; My Cart …

[PDF] mdpi.com

Bacterial outer membrane vesicles (omvs)-based dual vaccine for influenza a h1n1 virus and mers-cov

MM Shehata, A Mostafa, L Teubner, SH Mahmoud… - Vaccines, 2019 - mdpi.com

Vaccination is the most functional medical intervention to prophylactically control severe diseases
caused by human-to-human or animal-to-human transmissible viral pathogens. Annually, seasonal
influenza epidemics attack human populations leading to 290–650 thousand deaths …

[HTML] nih.govFull View

[HTML] Engineered outer membrane vesicle is potent to elicit HPV16E7-specific cellular immunity in a mouse model of TC-1 graft tumor

S Wang, W Huang, K Li, Y Yao, X Yang… - International journal …, 2017 - ncbi.nlm.nih.gov

… Second, the ability of recombinant OMVs delivering their components and the model antigen
green fluorescent protein to antigen-presenting cells was investigated in the macrophage
Raw264.7 cells and in bone marrow-derived dendritic cells in vitro …

Цитируется: 8 Похожие статьи Все версии статьи (6) Web of Science: 7

[HTML] discoverymedicine.com

[HTML] Gram-negative outer membrane vesicles in vaccine development

BS Collins - Discovery medicine, 2011 - discoverymedicine.com

… For instance, Schroeder and Aebischer (2009) prepared recombinant OMVs from
Salmonella carrying Leishmania antigens fused to C-terminal domains of an E. coli
autotransporter that spontaneously integrates into the OM …

[PDF] pnas.orgFree from Publisher

Bacterial outer membrane vesicles engineered with lipidated antigens as a platform for Staphylococcus aureus vaccine

C Irene, L Fantappiè, E Caproni… - Proceedings of the …, 2019 - National Acad Sciences

Skip to main content. Submit; About: Editorial Board; PNAS Staff; FAQ; Rights and Permissions;
Site Map. Contact; Journal Club; Subscribe: Subscription Rates; Subscriptions FAQ; Open Access;
Recommend PNAS to Your Librarian. Log in; Log out; My Cart. Main menu …

The difficulty with such an approach to a vaccine is the batch-to-batch variation plus the laborious, potentially hazardous microbiological processes involved.

Despite the complexity of this issue, it was successfully resolved in relation to OMVs Neisseria meningitidis type B vaccine. Meningococcal group B outer membrane vesicle vaccines have been used widely in Cuba, New Zealand, and Brazil [DOI: 10.1080/21645515.2017.1381810 ].

Additional questions/comments:

Point 2: abstract line 25: 'capability' should be 'ability'

Response 2: We have made a correction.

Point 3: Introduction lines 51-54: 'sub-unit vaccines mainly induce a short-lived humoral response': this depends on the formulation and excipients, a fact which should be mentioned. see Williamson et al Infect Immun 2005, 73, 3598-3608 for human immune response to sub-unit vaccine out to 3 months with evidence of functional antibody at this timepoint; also Williamson et al 2007 which showed that functional antibody was maintained in macaques for >1year, without further boosting ( Microb Pathogenesis 42, 12-22).

Response 3: Yes, indeed, there are published results that directly indicate the prolonged circulation of antibodies in the sera of experimental animals and people who were immunized with subunit vaccines. In this regard, we have corrected the text in accordance with the comment (lines 54-60).

Point 4: Line 62: 'Both the....'

Response 4: We have made a correction.

Point 5: line 68: 'protectivity' should be 'protective efficacy'

Response 5: We have made a correction.

Point 6: Line 128 and Table 1: these are all native proteins: that should be made clear in the table title

Response 6: We have made a correction.

Point 7: Lines 151-152 In my own experience guinea pigs are not a good model for plague. It may be unethical to suggest that more species are needed in vaccine efficacy testing -certainly species such as baboons. See also lines 273-4: essential to show efficacy in guinea pigs as well as mice?  how about an appropriate NHP species (e.g. macaque?)

Response 7: The choice of laboratory or wild animals for studying plague pathogenesis and immunogenesis can become the subject of a separate review article (if you wish, we can prepare it together). Guinea pigs are a classic model for plague and in our own experience guinea pigs are an excellent model for plague. In Russia, the following sequence of evaluating plague vaccines in preclinical trials is fixed: mice -> guinea pigs -> non-human monkeys (Papio hamadryas). In Russia, the use of certain animal species for conducting preclinical trials of candidate plague vaccines is regulated by [Main requirements for vaccine strains of the plague pathogen: Methodological Guidelines MU 3.3.1.1113-02] in Russian. DOI: 10.13140/RG.2.1.1468.6246

In other countries, guinea pigs are also in use:

An aroA mutant of Yersinia pestis is attenuated in guinea-pigs, but virulent in mice

PCF Oyston, P Russell, ED Williamson… - …, 1996 - microbiologyresearch.org

… the disease in humans, mice and guinea-pigs were also identified as being susceptible to Y.
pestis infection and … GBAaroA was attenuated in guinea-pigs in vivo … The virulence of GBAaroA
in mice contrasts with the behaviour of a AaroA mutant of Y. enterocolitica, which was not …

Studies on immunization against plague. V. Multiplication and persistence of virulent and avirulent Pasteurella pestis in mice and guinea pigs

DL Walker, LE Foster, TH Chen… - The Journal of …, 1953 - Am Assoc Immnol

Some avirulent strains of Pasteurella pestis progressively decrease in numbers in host
tissue without indication of multiplication, but viable bacilli can still be found up to the 6th to
the 9th days after inoculation. Other strains maintain their numbers well for 5 or 6 days and …

[PDF] asm.org

Pathogenicity and immunogenic efficacy of a live attenuated plague vaccine in vervet monkeys

AF Hallett, M Isaäcson, KF Meyer - Infection and immunity, 1973 - Am Soc Microbiol

… Experimental work on live attenuated plague vaccines using genetically related Y. pestis strains
established the importance of virulence determinants in immunity to plague in white mice, guinea
pigs, and rabbits (3, 4). The strains which were most immunogenic in …

Point 8: Lines 334-341: what is the 'protective substance' referred to here from Y.pseudotuberculosis? These lines are difficult to follow....

Response 8: We have now indicated that this is an unidentified protective substance and described the way of its isolation.

Point 9: Lines 419: Babbons...'did not feel sick'. How did these authors know (unless baboons can talk?). this is an anthropomorphic comment and should be removed.

Response 9: We wrote "look sick" instead of "feel sick".

Point 10: Lines 420-425: no doubt cellular immunity is required (both humoral and cellular immunity are required for protection) but the cellular component is overstated here

Response 10: Indeed, during immunization of mice, humoral immunity prevails. But when working with guinea pigs (which the reviewer considers a poor model for plague), the experimenter finds a certain percentage of seronegative animals that are nonetheless resistant to plague infection. Seronegative individuals are also found among immunized monkeys and humans.

Point 11: Line 526: why is it 'obviously'? EV vaccines are efficacious but as mentioned previously in this review can cause severe side effects. Sub-unit vaccines (depending on formulation) can provide a safe and efficacious alternative and can be produced in large consistent quantity without hazard. Please can the authors present a more balanced argument. Then the case for OMV's can be assessed more effectively.

Response 11: In order to make our position clear, we specified in the text that subunit vaccines based on F1 and/or V antigens protect mice, when infected with "classical" Y. pestis strains, are not worse than the live plague vaccine, but significantly inferior when mice were infected with F1-negative variants and/or bacteria producing isoforms of V antigen differing from antigens used for immunization. In addition, subunit vaccines are inferior to live vaccines in protection induced by immunization of guinea pigs and some monkey species.

Comments and Suggestions for Authors

Point 1: This is a comprehensive review, but rather long. I suggest this review could be considerably shortened without loss of impact.

Response 1: Thank you for your careful reading of the manuscript and helpful comments.

Yes, the manuscript is indeed quite long. This is due to the fact that it includes a review of not only readily available English-language sources, but also a number of Russian-language publications that are inaccessible to the vast majority of readers, but include data from rather expensive experiments using guinea pigs and non-human monkeys, as well as an alternative logic of experimental research aimed to develop an effective plague vaccine. We tried to shorten the text of the manuscript whenever possible, but the answers to the questions and comments of the reviewers practically canceled our efforts.

My major question about the OMV approach is: how do the authors propose to make it scalable?

The OMV approach is already scalable. Lines 308-311: More than 30 million people have been immunized with OMVs Neisseria meningitidis type B vaccine … [103].

could it ever become a recombinant process?

OMVs experimental recombinant vaccines against a number of bacterial pathogens have been developed and tested on laboratory rodents:

Antibody-mediated immunity induced by engineered Escherichia coli OMVs carrying heterologous antigens in their lumen

L Fantappiè, M De Santis, E Chiarot… - Journal of …, 2014 - Taylor & Francis

… As shown in Fig. 5, the 3 recombinant OMVs induced antigen-specific antibody responses which,
in the case of Slo dm -OMVs and SpyCEP-OMVs, were in the same range as the titers elicited
by 20 µg doses of the corresponding recombinant antigens formulated in Alum …

[PDF] tandfonline.comFull View

Recombinant outer membrane vesicles carrying Chlamydia muridarum HtrA induce antibodies that neutralize chlamydial infection in vitro

E Bartolini, E Ianni, E Frigimelica… - Journal of …, 2013 - Taylor & Francis

… To the best of the authors' knowledge, this is the first demonstration that, by virtue of their ability
to carry properly folded membrane-associated proteins, recombinant OMVs can be exploited
to induce functional immune responses that otherwise would be difficult to elicit …

[HTML] plos.orgFull View

[HTML] Mechanistic insight into the TH 1-biased immune response to recombinant subunit vaccines delivered by probiotic bacteria-derived outer membrane vesicles

JA Rosenthal, CJ Huang, AM Doody, T Leung… - PloS one, 2014 - journals.plos.org

… Noting that the untapped potential of recombinant OMVs aligned well with key limitations of subunit
vaccines, we investigated whether a T H 1-biased recombinant vaccine delivery platform could
be created from OMVs derived from Gram-negative probiotic bacteria …

Recombinant protein meningococcal serogroup B vaccine combined with outer membrane vesicles

X Bai, J Findlow, R Borrow - Expert opinion on biological therapy, 2011 - Taylor & Francis

… Several approaches have been taken to increase the breadth of coverage of OMV vaccines. One
has been the use of recombinant OMVs to express multiple PorA antigens and led to the
development of HexaMen and NonaMen by the Netherlands Vaccine Institute …

[PDF] pnas.orgFree from Publisher

Delivery of foreign antigens by engineered outer membrane vesicle vaccines

DJ Chen, N Osterrieder, SM Metzger… - Proceedings of the …, 2010 - National Acad Sciences

Skip to main content. Submit; About: Editorial Board; PNAS Staff; FAQ; Accessibility Statement;
Rights and Permissions; Site Map. Contact; Journal Club; Subscribe: Subscription Rates;
Subscriptions FAQ; Open Access; Recommend PNAS to Your Librarian. Log in; Log out; My Cart …

[PDF] mdpi.com

Bacterial outer membrane vesicles (omvs)-based dual vaccine for influenza a h1n1 virus and mers-cov

MM Shehata, A Mostafa, L Teubner, SH Mahmoud… - Vaccines, 2019 - mdpi.com

Vaccination is the most functional medical intervention to prophylactically control severe diseases
caused by human-to-human or animal-to-human transmissible viral pathogens. Annually, seasonal
influenza epidemics attack human populations leading to 290–650 thousand deaths …

[HTML] nih.govFull View

[HTML] Engineered outer membrane vesicle is potent to elicit HPV16E7-specific cellular immunity in a mouse model of TC-1 graft tumor

S Wang, W Huang, K Li, Y Yao, X Yang… - International journal …, 2017 - ncbi.nlm.nih.gov

… Second, the ability of recombinant OMVs delivering their components and the model antigen
green fluorescent protein to antigen-presenting cells was investigated in the macrophage
Raw264.7 cells and in bone marrow-derived dendritic cells in vitro …

Цитируется: 8 Похожие статьи Все версии статьи (6) Web of Science: 7

[HTML] discoverymedicine.com

[HTML] Gram-negative outer membrane vesicles in vaccine development

BS Collins - Discovery medicine, 2011 - discoverymedicine.com

… For instance, Schroeder and Aebischer (2009) prepared recombinant OMVs from
Salmonella carrying Leishmania antigens fused to C-terminal domains of an E. coli
autotransporter that spontaneously integrates into the OM …

[PDF] pnas.orgFree from Publisher

Bacterial outer membrane vesicles engineered with lipidated antigens as a platform for Staphylococcus aureus vaccine

C Irene, L Fantappiè, E Caproni… - Proceedings of the …, 2019 - National Acad Sciences

Skip to main content. Submit; About: Editorial Board; PNAS Staff; FAQ; Rights and Permissions;
Site Map. Contact; Journal Club; Subscribe: Subscription Rates; Subscriptions FAQ; Open Access;
Recommend PNAS to Your Librarian. Log in; Log out; My Cart. Main menu …

The difficulty with such an approach to a vaccine is the batch-to-batch variation plus the laborious, potentially hazardous microbiological processes involved.

Despite the complexity of this issue, it was successfully resolved in relation to OMVs Neisseria meningitidis type B vaccine. Meningococcal group B outer membrane vesicle vaccines have been used widely in Cuba, New Zealand, and Brazil [DOI: 10.1080/21645515.2017.1381810 ].

Additional questions/comments:

Point 2: abstract line 25: 'capability' should be 'ability'

Response 2: We have made a correction.

Point 3: Introduction lines 51-54: 'sub-unit vaccines mainly induce a short-lived humoral response': this depends on the formulation and excipients, a fact which should be mentioned. see Williamson et al Infect Immun 2005, 73, 3598-3608 for human immune response to sub-unit vaccine out to 3 months with evidence of functional antibody at this timepoint; also Williamson et al 2007 which showed that functional antibody was maintained in macaques for >1year, without further boosting ( Microb Pathogenesis 42, 12-22).

Response 3: Yes, indeed, there are published results that directly indicate the prolonged circulation of antibodies in the sera of experimental animals and people who were immunized with subunit vaccines. In this regard, we have corrected the text in accordance with the comment (lines 54-60).

Point 4: Line 62: 'Both the....'

Response 4: We have made a correction.

Point 5: line 68: 'protectivity' should be 'protective efficacy'

Response 5: We have made a correction.

Point 6: Line 128 and Table 1: these are all native proteins: that should be made clear in the table title

Response 6: We have made a correction.

Point 7: Lines 151-152 In my own experience guinea pigs are not a good model for plague. It may be unethical to suggest that more species are needed in vaccine efficacy testing -certainly species such as baboons. See also lines 273-4: essential to show efficacy in guinea pigs as well as mice?  how about an appropriate NHP species (e.g. macaque?)

Response 7: The choice of laboratory or wild animals for studying plague pathogenesis and immunogenesis can become the subject of a separate review article (if you wish, we can prepare it together). Guinea pigs are a classic model for plague and in our own experience guinea pigs are an excellent model for plague. In Russia, the following sequence of evaluating plague vaccines in preclinical trials is fixed: mice -> guinea pigs -> non-human monkeys (Papio hamadryas). In Russia, the use of certain animal species for conducting preclinical trials of candidate plague vaccines is regulated by [Main requirements for vaccine strains of the plague pathogen: Methodological Guidelines MU 3.3.1.1113-02] in Russian. DOI: 10.13140/RG.2.1.1468.6246

In other countries, guinea pigs are also in use:

An aroA mutant of Yersinia pestis is attenuated in guinea-pigs, but virulent in mice

PCF Oyston, P Russell, ED Williamson… - …, 1996 - microbiologyresearch.org

… the disease in humans, mice and guinea-pigs were also identified as being susceptible to Y.
pestis infection and … GBAaroA was attenuated in guinea-pigs in vivo … The virulence of GBAaroA
in mice contrasts with the behaviour of a AaroA mutant of Y. enterocolitica, which was not …

Studies on immunization against plague. V. Multiplication and persistence of virulent and avirulent Pasteurella pestis in mice and guinea pigs

DL Walker, LE Foster, TH Chen… - The Journal of …, 1953 - Am Assoc Immnol

Some avirulent strains of Pasteurella pestis progressively decrease in numbers in host
tissue without indication of multiplication, but viable bacilli can still be found up to the 6th to
the 9th days after inoculation. Other strains maintain their numbers well for 5 or 6 days and …

[PDF] asm.org

Pathogenicity and immunogenic efficacy of a live attenuated plague vaccine in vervet monkeys

AF Hallett, M Isaäcson, KF Meyer - Infection and immunity, 1973 - Am Soc Microbiol

… Experimental work on live attenuated plague vaccines using genetically related Y. pestis strains
established the importance of virulence determinants in immunity to plague in white mice, guinea
pigs, and rabbits (3, 4). The strains which were most immunogenic in …

Point 8: Lines 334-341: what is the 'protective substance' referred to here from Y.pseudotuberculosis? These lines are difficult to follow....

Response 8: We have now indicated that this is an unidentified protective substance and described the way of its isolation.

Point 9: Lines 419: Babbons...'did not feel sick'. How did these authors know (unless baboons can talk?). this is an anthropomorphic comment and should be removed.

Response 9: We wrote "look sick" instead of "feel sick".

Point 10: Lines 420-425: no doubt cellular immunity is required (both humoral and cellular immunity are required for protection) but the cellular component is overstated here

Response 10: Indeed, during immunization of mice, humoral immunity prevails. But when working with guinea pigs (which the reviewer considers a poor model for plague), the experimenter finds a certain percentage of seronegative animals that are nonetheless resistant to plague infection. Seronegative individuals are also found among immunized monkeys and humans.

Point 11: Line 526: why is it 'obviously'? EV vaccines are efficacious but as mentioned previously in this review can cause severe side effects. Sub-unit vaccines (depending on formulation) can provide a safe and efficacious alternative and can be produced in large consistent quantity without hazard. Please can the authors present a more balanced argument. Then the case for OMV's can be assessed more effectively.

Response 11: In order to make our position clear, we specified in the text that subunit vaccines based on F1 and/or V antigens protect mice, when infected with "classical" Y. pestis strains, are not worse than the live plague vaccine, but significantly inferior when mice were infected with F1-negative variants and/or bacteria producing isoforms of V antigen differing from antigens used for immunization. In addition, subunit vaccines are inferior to live vaccines in protection induced by immunization of guinea pigs and some monkey species.

Round 2

Reviewer 2 Report

need editing of English language